# One Health Approach: Antibiotic Resistance Among Enterococcal Isolates in Dairy Farms in Selangor

**DOI:** 10.3390/antibiotics14040380

**Published:** 2025-04-04

**Authors:** Sakshaleni Rajendiran, Yuvaneswary Veloo, Salina Abdul Rahman, Rohaida Ismail, Zunita Zakaria, Rozaihan Mansor, Maslina Mohd Ali, Hassuzana Khalil, Syahidiah Syed Abu Thahir

**Affiliations:** 1Environmental Health Research Centre, Institute for Medical Research, Ministry of Health Malaysia, Shah Alam 40170, Malaysia; rohaidadr@moh.gov.my (R.I.); syahidiah@moh.gov.my (S.S.A.T.); 2Nutrition, Metabolic and Cardiovascular Research Centre (NMCRC), Institute for Medical Research, Ministry of Health Malaysia, Shah Alam 40170, Malaysia; sar@moh.gov.my; 3Institute for Bioscience, Universiti Putra Malaysia, Serdang 43400, Malaysia; zunita@upm.edu.my (Z.Z.); rozaihan@upm.edu.my (R.M.); 4Faculty Veterinary Medicine, Universiti Putra Malaysia, Serdang 43400, Malaysia; 5Department of Veterinary Services Selangor, Shah Alam 40200, Malaysia; maslina@dvs.gov.my (M.M.A.); hassuzana@dvs.gov.my (H.K.)

**Keywords:** enterococci, dairy farm, antibiotic resistance, workers, milk, environment

## Abstract

Background/Objectives: Antibiotic resistance is a growing public health concern. The One Health approach is essential in addressing antibiotic (AR) resistance. Therefore, this study aimed to determine AR among enterococcal isolates in dairy farms across various domains based on the emerging dairy industry. Methods: A total of 208 enterococcal isolates from the workers = 70, milk = 76, and environment = 62 of eight dairy farms in Selangor, Malaysia, were used in this study. The bacterial identification and antibiotic susceptibility testing (AST) were conducted utilising the Vitek-2 system. Results: Approximately 28% of the isolates exhibited susceptibility to all tested antibiotics. A relatively higher proportion of isolates demonstrated resistance to tetracycline, which was followed by erythromycin. The multidrug resistance (MDR) and multiple antibiotic resistance (MAR) index were low in this study. Conclusions: The studied dairy farms exhibited lower AR, MDR, and MAR index values. Nevertheless, ongoing surveillance is essential due to the recent expansion of the dairy farming industry.

## 1. Introduction

Antibiotic resistance (AR) is an increasing concern that has attracted global public health attention. This phenomenon has resulted in numerous initiatives currently being executed through the collaboration of international organisations. Notably, the One Health approach is essential for addressing AR by integrating human, animal, and environmental health [1]. Approximately 75% of human infectious diseases over the past decades have also been estimated to have originated from animals, underscoring the significance of the One Health approach in tackling AR [2]. Furthermore, the transmission and spread of resistant bacteria are influenced by multiple factors. Nevertheless, the primary drivers of this phenomenon are the use and overuse of antibiotics in healthcare settings and the livestock industry [2,3].

Given the decline in new antibiotic discoveries, implementing all potential human and veterinary medicine measures is necessary to safeguard the existing antimicrobial arsenal from further resistance [3]. The World Organisation for Animal Health has classified antimicrobial agents critical to veterinary medicine, emphasising their prudent use for animal health and productivity. Each antibiotic possesses a specific indication and species [4]. A review has also identified the significant use of antibiotics among dairy and swine farmers for mastitis and weaning [5]. Even though using antibiotics in animal feed as a growth promoter raises concerns regarding animal health, numerous countries have banned, restricted, or phased out this practice. The countries include Europe, the United States, Australia, Korea, China, India, Thailand, and Malaysia [6,7,8].

The rising global demand for protein has drawn attention to South Asia as a potential hub for the livestock industry [9]. Southeast Asian countries have also exhibited a significant imprudent antibiotic use while serving as a focal point for AR [10]. Thus, the application of antibiotics in industries that promote growth, prophylaxis, and treatment to enhance productivity can further exacerbate the spread of AR. In addition, using antibiotics in livestock animals can pose a risk to humans through occupational exposure to consumers and nearby communities [11].

The feed production in Malaysia, for ruminants, is far less than that of other farming industries. Most farmers within the ruminant category (dairy farmers) also primarily operate small-scale farms or semi-intensive systems [12]. On the contrary, commercial operations with intensive production systems are practised in poultry and swine farming [12]. This observation accounts for the prominence of these fields in AR-related studies. One notable example is the One Health approach. Nonetheless, insufficient information on dairy farming has been noted. Various measures have been implemented and are ongoing to enhance this industry due to the milk production being significantly below the self-sufficiency level (SSL). Among those measures, the establishment of milk collection centres is aimed at supporting small-scale farmers [13].

This study addressed the gap in information regarding AR in the dairy farming industry in Selangor, Malaysia, on the One Health approach to provide baseline data. The selection of specific indicator bacterial strains among diverse species was crucial for determining the distribution of AR, which could substantially influence policy decisions regarding the prudent use of antibiotics in veterinary practice [3]. Typically, enterococci (Gram-positive bacteria) are widely distributed and serve as essential indicator bacteria for resistance in human and veterinary surveillance systems [14]. *Enterococcus* spp. are also acknowledged as important species in the food production industry for AR and for their role in transmitting human infections [15]. Therefore, this study assessed the AR distribution among enterococcal isolates in dairy farms in Selangor, Malaysia, across multiple domains (workers, milk, and the environment).

## 2. Results

### 2.1. Distribution of Enterococcal Isolates

Among 208 enterococcal isolates recovered from eight dairy farms in Selangor, the predominant species was *Enterococcus faecalis* (*E. faecalis*, n = 119, 57%). This finding was followed by *Enterococcus gallinarum* (*E. gallinarum*, n = 30, 14%), *Enterococcus casseliflavus* (*E. casseliflavus*, n = 26, 13%), *Enterococcus faecium* (*E. faecium*, n = 23, 11%), and others (n = 10, 5%). The prevalence of *E. casseliflavus*, *E. gallinarium*, and other enterococcal species in the environment was also more significant than in the different domains, leading to their classification as other enterococcal species in this study. Likewise, the distribution of *Enterococcus* spp. exhibited substantial variation across the domain and farms (*p* < 0.01) in which *E. faecalis* and *E. faecium* were predominantly observed in milk and workers, respectively (Figure 1). Nevertheless, no statistical differences were observed between *Enterococcus* spp. and farm types (*p* > 0.05).

### 2.2. Antibiotic-Susceptible and -Resistant Isolates

Approximately 100 isolates (48%) demonstrated susceptibility to all tested antibiotics, with the majority originating from the environment (n = 34, 55%) and humans (n = 38, 54%). These outcomes were followed by milk (n = 28, 37%). Statistically significant differences were also computed in the enterococcal isolates that were susceptible to all tested antibiotics across different domains, farms, and farm types. Meanwhile, a higher proportion of isolates revealing susceptibility to all tested antibiotics was obtained from large-scale farms (67%). This result was followed by semi-commercial farms (52%). All isolates demonstrated susceptibility to tigecycline, while a higher proportion of isolates exhibited resistance to tetracycline (n = 89, 43%). This observation was accompanied by a predominant occurrence in small-scale farms (n = 40, 95%). Significant differences were observed among isolates exhibiting resistance towards erythromycin (*p* = 0.01) across the domain. These isolates were primarily observed in farmers. Moreover, notable differences were observed among isolates resistant to streptomycin (*p* < 0.01) and erythromycin (*p* < 0.01) across farm types, which were predominantly in small-scale farms. Table 1 shows the AR distribution observed among enterococcal isolates categorised by domains and species.

About 43 enterococcal isolates exhibited resistance towards two or more tested antibiotics, identifying 17 distinct AR profiles. Table 2 shows the AR profiles of isolates exhibiting resistance to three or more antibiotics. The phenotypes of P1 and P4 revealed the highest prevalence in the AR profile of *E. faecalis*. Linezolid-resistant enterococci (LRE) were also determined among isolates with multiple resistance. All LRE were associated with resistance to tetracycline and erythromycin. In contrast, none of the vancomycin-resistant enterococci (VRE) exhibited co-resistance to linezolid. The environment domain containing VRE (n = 2) and LRE (n = 2) was then isolated from the effluent water of a commercial dairy farm. Likewise, the VRE from milk samples (n = 2) originated from small and semi-commercial farms. Despite LRE isolates being obtained from milk from a large farm, the small-scale farm exclusively employed farmers with LRE.

### 2.3. Multidrug Resistance

Of the 208 enterococcal isolates obtained from dairy farms, 5% (n = 11) demonstrated multidrug resistance (MDR). The distribution across domains is as follows: human (n = 3), milk (n = 4), and the environment (n = 4). A higher proportion of enterococcal isolates exhibited resistance to four antibiotic classes (n = 7), with predominant occurrences in milk (n = 3) and the environment (n = 3). These isolates were commonly found in commercial farms (n = 6). Four isolates also revealed resistance to three antibiotic classes primarily associated with farmers (n = 2).

### 2.4. Multiple Antibiotic Resistance Index

The multiple antibiotic resistance (MAR) index of enterococcal isolates recovered from dairy farms in Selangor ranged from 0 to 0.58. Approximately 14% of enterococcal isolates (n = 30) demonstrated a MAR index of 0.2 or higher, which implies exposure to high-risk sources of antibiotic contamination. Particularly, the maximum MAR index value of 0.58 was observed in three isolates from milk samples (Figure 2). Significant differences in the high MAR index of isolates were also noted across domains (*p* < 0.001), with a predominance observed among workers (n = 20, 29%). Small-scale farms with reduced herd sizes then exhibited a higher proportion of isolates with a high MAR index (n = 22, 27%). The MAR index of those isolates ranged from 0.2 to 0.33. Alternatively, all isolates recovered from semi-commercial farms revealed a low MAR index. The proportions of isolates with high MAR indices in commercial and large-scale farms were 17% and 7%, respectively. Notably, the high MAR index for commercial farms was between 0.2 and 0.5. The isolates with the highest MAR index value of 0.58 were recovered from large-scale farms.

## 3. Discussion

This study indicated that the distribution of enterococci varied among the domains, and a higher number of enterococcal isolates were recovered from milk samples. Enterococci in milk could originate from the milking machines and bulk tanks [16]. Nonetheless, *E. faecalis* and *E. faecium* strains held paramount importance from the lens of public health [17,18]. This study also observed that *E. faecalis* isolates were predominantly detected in milk samples, with a subsequent distribution rate among farmers. The *E. faecalis* was predominantly recorded in raw milk [16]. On the contrary, *E. faecium* exhibited a relatively lower distribution rate in the studied dairy farms. This species was identified in the ESKAPE (*Enterococcus faecium*, *Staphylococcus aureus*, *Klebsiella pneumoniae*, *Acinetobacter baumannii*, *Pseudomonas aeruginosa*, *Enterobacter* spp.) in the global priority pathogen list [19].

The Malaysian Veterinary Antibiotic Guidelines (MVAGs) were initially released in 2021 for veterinary practitioners. The document listed veterinary important antimicrobials detailing their use, indications, dosage, target species, duration, withdrawal period, and maximum residue limit [20]. Typically, tetracycline is categorised as a veterinary critically important antimicrobial (VCIA) and is also recognised as a highly important antimicrobial in human health [20,21]. This antibiotic is utilised in cattle for treating various infections [20], accounting for the comparatively higher proportions of enterococcal isolates in this study that exhibited resistance. Alterations in efflux mechanisms or ribosomal protection also frequently account for the resistance [22]. Interestingly, this study implied that none of the enterococcal isolates exhibited resistance to tigecycline, which was designed to address these resistance mechanisms [22].

Erythromycin exhibited a comparatively higher level of resistance than the other tested antibiotics. Generally, this antibiotic is classified as a macrolide and is utilised for various livestock industry applications. This antibiotic is also recognised as a VCIA and a critically important antimicrobial in human medicine [20,21]. Although enterococci are intrinsically susceptible to tetracycline and erythromycin [17], the higher resistance observed in small-scale farms in this study suggested the potential use of these antibiotics in such settings [18]. Moreover, the ban on those antibiotics in Malaysian livestock feed in 2020 [23] provided a framework for this study to analyse subsequent trends.

The prohibition of avoparcin and vancomycin in veterinary medicine has been correlated with a subsequent reduction in the prevalence of VRE resistance within the livestock industry [14]. Previous studies on local livestock industries also reported a similar phenomenon [8,24,25]. Consequently, the rate of VRE (1%) was relatively low in this study. Nevertheless, the presence of VRE in milk and environmental samples in this study was contentious. Even though the proportion of VRE in milk was observed to be lower than in previous research [26], other studies reported that all recovered enterococcal isolates exhibited susceptibility to vancomycin [27,28]. The VRE were also isolated from farms of varying scales in this study. Considering that vancomycin is one of the last-resort treatments for enterococcal infection in clinical settings [8], this finding signified the need to examine the VRE concerning its occurrence in milk and farm effluent. Furthermore, the VRE from milk that enters the consumer’s gut are usually carried in the intestine. Consumers harbouring VRE may act as reservoirs, disseminating it within the community and environment [29].

Similar to vancomycin, linezolid is regarded as one of the last-resort options for enterococcal infections in human medicine [30]. Moreover, it is categorised as a critically important antimicrobial for human medicine and is not listed in the use of veterinary medicine [31]. This study reported that LRE was 4% and distributed across all domains. On the contrary, a prevalence of 1.6% for LRE was reported in the clinical surveillance within Malaysia [32]. A systematic review and meta-analysis conducted on LRE globally also concluded a pooled prevalence of 3.3%, with human prevalence at 1.9% and animal prevalence at 6.3%. Although this value was relatively low, the scenario remained a concern [33]. Thus, monitoring linezolid or any other agents with similar properties in the food production industry is pivotal. Moreover, this study identified LRE among isolates exhibiting multiple resistance. Plasmids containing mobile linezolid resistance genes were reported to carry co-resistance genes, especially phenicol resistance genes. Hence, analysing these genes from the farm environment is crucial for addressing dissemination [34]. Given that molecular analysis-related future studies can yield insights into resistance genes, they remain vital.

The rate of MDR among enterococcal isolates was relatively low in this study. Considering that all the farms were registered under the Department of Veterinary Services (DVS), Selangor, it was conjectured that the surveillance and awareness programmes could contribute to this outcome. Therefore, addressing farms not registered in the DVS in future studies is vital, as it could reveal differing trends in antibiotic usage and surveillance. The MDR isolates primarily obtained from commercial farms also raised concerns regarding the implications of antimicrobial usage. The higher MDR rate in milk samples and the environment posed concerns about the transmission of AR to humans, whether through consumption or direct or indirect environmental contact. Additionally, an environment containing MDR bacteria may serve as a source of re-infection among farm cattle. The MDR enterococcal isolates can also horizontally transfer genes to other species within the same environment. This process can exacerbate the phenomenon in the near future [35]. Hence, enhancing biosecurity practices is essential to mitigate the spread of resistant bacteria within and between farms.

The MAR index is a screening tool for identifying the source of antibiotic contamination. Compared to other food-producing industries in this nation, the MAR index of the isolates in this study was relatively low [15,36,37]. However, the proportion of high MAR index values was relatively higher in small-scale farms and commercial farms. Conversely, none of the isolates recovered from semi-commercial farms revealed a high MAR index. The discrepancy of MAR index discrepancies among the farms implies that there might be variations in the biosecurity practices, farm management, and antibiotic usage. Even though the proportion of isolates with a high MAR index was lower in large-scale farms, the highest MAR index value was reported in milk samples. This outcome implies that there is a high-risk source of contamination with AR in milk samples, meaning that milk consumption could pose a threat to consumers through the potential dissemination of AR genes [38]. Thus, this study recommended consuming milk after boiling or pasteurisation.

Nevertheless, the high MAR index was primarily found among workers on small farms. Dairy farmers countered risks associated with microbial exposure, including bioaerosols that could harbour microbial agents [39]. This phenomenon suggests increased personal protective equipment usage among the farmers to safeguard against microbial exposure [40]. Moreover, it is imperative to educate the farmers on personal protection and personal hygiene. One prominent example is attention to proper handwashing techniques using soaps, which farmers often neglect. These dairy farmers who are directly or indirectly exposed to AR bacteria can transmit them to the community and animals [41]. The availability of data regarding biosecurity practices on farms and recent antibiotic usage among the farmers was also a limitation of this study.

This study examined the domains of humans, milk, and the environment through the One Health approach to determine the AR distribution in dairy farms. The initiation of food safety began at the farm and could be transmitted to the community via workers, food products, and the environment. Therefore, the One Health approach is essential for the dairy farm industry [42]. A coordinated surveillance system involving multiple agencies, including public health, veterinary, and environmental health sectors, is also critical for addressing this issue. The effort can focus on monitoring resistant patterns and trends while determining potential hotspots for resistance transmission. Hence, a targeted intervention can be executed. Furthermore, monitoring resistance in the food production industry can facilitate the assessment and comparison of clinical trends in future studies. This suggestion highlights the necessity of data sharing with stakeholders to enable early detection and response to emerging resistance threats.

Given the discrepancies among studies regarding the selection of antibiotics in assessing resistance rates, this study focuses on important antibiotics relevant to human medicine [18,28,43]. Certain antibiotics are crucial for treating severe enterococcal infection in humans, and resistance to these antibiotics is challenging to the treating physician. The findings from this study could also contribute to national and international AR monitoring efforts by emphasising clinically important antibiotics. Meanwhile, the national guidelines have primarily focused on AR data from hospitals concerning human health. The implementation of AR surveillance for dairy cattle in Malaysia commenced in 2021 [44]. Hence, including samples from the environment and workers in the surveillance programmes is essential. Furthermore, the distribution of AR may provide baseline information for certain antibiotics that were prohibited before the initiation of this study.

The study was limited by selection bias, as it focused on farms registered under DVS. Consequently, the findings could be biased due to the surveillance of DVS on those farms. This study also did not gather data on antibiotic use and farm management, which was an additional limitation. Moreover, this study did not conduct molecular analysis and comprehensive phenotypic evaluation. This constraint could restrict resistance assessment to either a genetically determined phenomenon or a phenotypic adaptation, along with the factors contributing to co-resistance. Thus, future studies could address these limitations.

## 4. Materials and Methods

### 4.1. Study Design and Sample Sources

This cross-sectional study focused on enterococcal isolates obtained from eight dairy farms in Selangor. The analysis utilised samples from two larger projects approved by the National Medical Research Registry, Ministry of Malaysia (NMRR-20-3072-57763 and NMRR-20-2798-57759), encompassing multiple domains and bacterial species. The dairy farms varied from small- to large-scale farms and were randomly chosen from the registry list provided by DVS, Selangor. Table 3 presents data regarding the types and quantities of lactating cows across each farm. The farm scale was categorised as small, semi-commercial, commercial, and large [45].

This study collected 264 samples, comprising 72 from farmers (nasal and hand swabs), 48 from milk, and 144 from environmental sources (soil, effluent water, and cow dung) between January 2022 and December 2023. A total of 208 enterococcal isolates were recovered from the collected samples, comprising 70 from workers, 76 from milk, and 62 from the environment. Antimicrobial susceptibility testing (AST) was also conducted on all recovered enterococci at the Institute for Medical Laboratory (IMR).

### 4.2. Sample Sources and Collection

Nasal and hand swabs were obtained from 36 farmers by medical officers utilising sterile FLOQSwabs^®^ (Copan, Brescia, Italy). These samples were transported into an eSwab^®^ tube of 1 mL of liquid Amies medium (Copan, Brescia, Italy). A sterile disposable bottle was also employed to collect 50 mL of unpasteurised milk from each bulk tank in the dairy farms. Likewise, approximately 200 mL of effluent samples were obtained using a long-handled stainless-steel ladle from six distinct areas of each farm, including drainage or pools of stagnant water within the farms. Meanwhile, soil samples weighing approximately 25 g were collected from six areas where the cows resided and grazed on each farm. The details of the method for collecting soil and effluent samples followed the methodology described in previous studies [8,46]. Six fresh cow dung samples were collected from each farm using sterile FLOQSwabs^®^ and transported into a FecalSwab™ tube containing 2 mL of modified Cary–Blair medium (Copan, Brescia, Italy). All samples were placed in sterile zip-lock plastic bags and transported to the Microbiology Laboratory, IMR, using a cool box.

### 4.3. Sample Preparation and Isolation of Enterococci

Fresh peptone water (Difco™, BD Diagnostics, New Jersey, USA) was prepared to conduct serial dilution for the bacterial isolation. Plastic bags and bottles containing effluent water and milk, respectively, were mixed through vertical and lateral shaking. The soil samples were also shaken and mixed with a spatula before weighing. Subsequently, 10 g of soil was transferred into a Falcon tube containing 90 mL of peptone water, constituting the first dilution. A similar approach was applied to 10 mL of milk and effluent water. Approximately 1 mL and 2 mL from eSwab^®^ and FecalSwab™ tubes were transferred into tubes containing 9 mL and 18 mL of peptone water, respectively. The initial dilution tubes were then vortexed. Subsequently, 1 mL from the initial dilution tubes was transferred to a second dilution tube containing 9 mL of peptone water. This process was repeated until a concentration of 10^−6^ was obtained. The tubes were vortexed following each fold of dilution.

Consequently, 1 mL from each final dilution tube was poured onto HiChrome *E. faecium* agar, a commercially prepared agar plate. The aliquot was then evenly dispersed on the agar using a sterile, disposable spreader and incubated at 37 °C for 24 h. Then, the colonies exhibited blue colouration for *E. faecalis* and green for *E. faecium* [47]. This study selected three representative colonies based on the colours of 30 to 300 isolates on the plate. Subcultures of each isolate were then obtained twice consecutively on Trypticase Soy Agar (TSA). This step was crucial for obtaining pure colonies before bacterial identification and AST.

### 4.4. Identification of Enterococcal Species and Antimicrobial Susceptibility Testing

Gram staining was performed to screen the isolates for Gram-positive cocci, either in pairs or short chains. Meanwhile, the Vitek 2 system was utilised for bacterial identification and AST. Further system details can be found in a previous study [8]. These isolates that met the criteria underwent bacterial identification using VITEK^®^2 Gram-Positive Identification cards (GP-ID) (bioMérieux, Nurtingen, Germany). Pure enterococcal colonies were inoculated into clear, 12 mm × 75 mm polystyrene test tubes containing 3 mL of Vitek^®^ 0.45% saline solution (bioMérieux, Nurtingen, Germany). This solution was mixed thoroughly until the turbidity reached 0.5 to 0.63 McFarland, which was measured using DensiCHEK™ Plus ((bioMérieux, Nurtingen, Germany).

The isolates identified as enterococci underwent AST using AST-GP67 cards (bioMérieux), which were designed for *Enterococcus* spp., *Staphylococcus* spp., and Streptococcus agalactiae [48]. A total of 21 different antibiotics were incorporated into the wells of this AST card. Only 12 antibiotics were intended for *Enterococcus* spp.: (1) penicillin: ampicillin (AMP) and benzylpenicillin (PEN); (2) fluroquinolones: ciprofloxacin (CIP) and levofloxacin (LVX); (3) macrolides: erythromycin (ERY); (4) aminoglycosides: high-level gentamicin (GEN) synergy and high-level streptomycin (STR) synergy; (5) oxazolidinones: linezolid (LZD); (6) nitrofuran: nitrofurantoin (NIT); (6) tetracyclines (TET); (7) glycylcyclines: tigecycline (TGC); and (8) glycopeptides: vancomycin (VAN).

The results were automatically generated by the system, referencing its database, and reported as minimum inhibitory concentration (MIC) values following the Global Clinical and Laboratory Standards Institute (CLSI) 2020 [8]. *E. faecalis* ATCC 29212 served as the control strain, and the Vitek outcomes were in alignment with the MICs of the quality control organism as specified in the package insert.

### 4.5. MAR Index

The MAR index serves as a tool for evaluating health risks by identifying the source of contamination [49,50]. The index for a single isolate was determined using the formula a/b. Particularly, the number of antibiotics to which each enterococcal isolate exhibited resistance was denoted as ‘a’. A total of 12 antibiotics tested against enterococci in this study were denoted as ‘b’. Typically, a MAR index of 0.2 and above implies a high-risk source of antibiotic contamination [50,51,52,53,54].

### 4.6. Statistical Analysis

Data were entered using Statistical Package for Social Science (SPSS) version 27.0 (IBM Corp., Armonk, NY, USA). The Python programming language was utilised in version 1.2.0 of the Google Colab platform for statistical analysis. Pandas and NumPy were primarily applied in data analytics. Graphs were plotted using Seaborn which is Python data visualisation library. chi2_contigency and fisher_exact functions were imported from the Scipy.stats sub-package. The chi-square test and Fisher’s exact test were employed to assess the significant differences between the groups of variables [55]. Notably, Fisher’s exact test was applied when the expected frequency was below 5 in over 20% of the cells [56]. A *p*-value less than 0.05 was deemed statistically significant.

## 5. Conclusions

This study indicated that the AR rate and MDR of enterococcal isolates were low. The resistance rate to vancomycin and linezolid was also relatively low. Nevertheless, these findings warranted attention from relevant stakeholders to enhance the existing surveillance efforts, considering that these antibiotics are the last resort for enterococcal-related infection. Most of the farms also displayed a low MAR index, implying a low-risk source of contamination with antibiotics or resistance genes. Nonetheless, scrutiny of LRE in dairy farms is crucial, as linezolid represents one of the last-resort antibiotics for clinical infections and is not approved for veterinary use. Given that dairy farming is an emerging industry, ongoing monitoring of antibiotic use and AR is crucial.

## Figures and Tables

**Figure 1 antibiotics-14-00380-f001:**
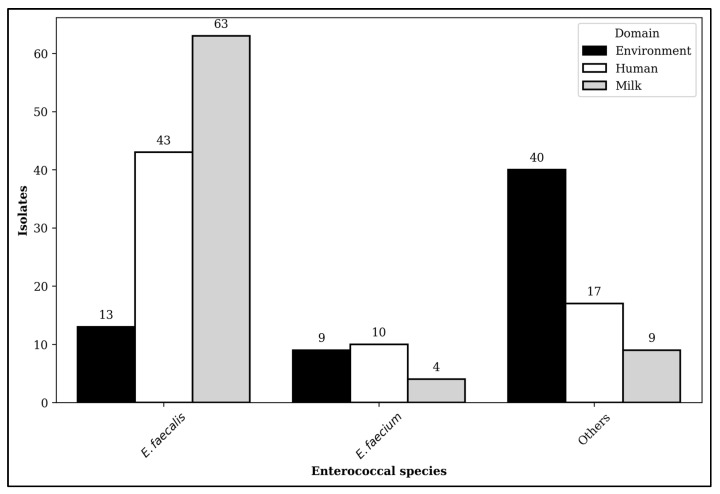
The distribution of enterococcal isolates recovered from dairy farms across domains.

**Figure 2 antibiotics-14-00380-f002:**
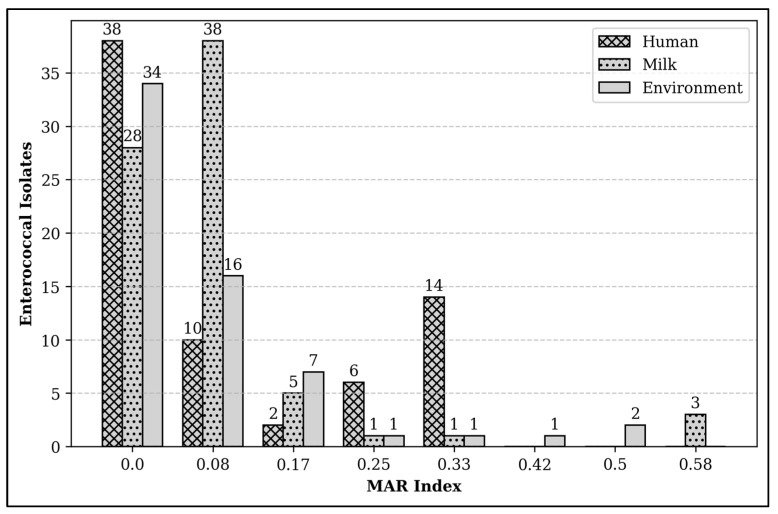
The MAR index of enterococcal isolates recovered from dairy farms by domain.

**Table 1 antibiotics-14-00380-t001:** Resistance rate of enterococcal isolates recovered from dairy farms in Selangor.

Antibiotic	Domain	Total Resistance, n (%)	Resistance by Species		
			*E. faecalis*, n (%)	*E. faecium*, n (%)	Others, n (%)
Ampicillin	Human	0	0	0	0
	Milk	0	0	0	0
	Environment	2 (3.2)	0	0	2 (5.0)
Benzylpenicillin	Human	0	0	0	0
	Milk	0	0	0	0
	Environment	2 (3.3)	0	0	2 (5.0)
Ciprofloxacin	Human	0	0	0	0
	Milk	4 (5.3)	4 (6.3)	0	0
	Environment	0	0	0	0
Erythromycin	Human	22 (31.4)	17 (39.5)	3 (30.0)	2 (11.8)
	Milk	9 (11.8)	8 (12.7)	0	1 (11.1)
	Environment	12 (19.4)	2 (15.4)	7 (77.8)	3 (7.5)
Gentamicin	Human	16 (22.9)	16 37.2)	0	0
	Milk	3 (3.9)	3 (4.8)	0	0
	Environment	0	0	0	0
Streptomycin	Human	15 (21.4)	15 (34.9)	0	0
	Milk	10 (13.2)	10 (15.9)	0	0
	Environment	12 (19.4)	5 (38.5)	3 (33.3)	4 (10.0)
Levofloxacin	Human	0	0	0	0
	Milk	4 (5.3)	4 (6.3)	0	0
	Environment	0	0	0	0
Linezolid	Human	3 (4.3)	0	1 (10.0)	2 (11.8)
	Milk	3 (3.9)	3 (4.8)	0	0
	Environment	2 (3.2)	0	0	2 (5.0)
Nitrofurantoin	Human	3 (4.3)	0	2 (20.0)	1 (5.9)
	Milk	1 (1.3)	1 (1.6)	0	0
	Environment	2 (1.3)	0	1 (11.1)	1 (2.5)
Tetracycline	Human	29 (41.4)	22 (51.2)	3 (30.0)	4 (30.0)
	Milk	40 (52.6)	37 (58.7)	0	3 (33.3)
	Environment	20 (32.3)	2 (15.4)	6 (66.7)	12 (30.0)
Vancomycin	Human	0	0	0	0
	Milk	2 (2.6)	1 (1.6)	0	1 (11.1)
	Environment	2 (3.2)	0	2 (22.2)	0

All *enterococcus* spp. exhibited susceptibility to tigecycline.

**Table 2 antibiotics-14-00380-t002:** Antibiotic resistance profiles of enterococcal isolates.

Number of Antibiotics	Phenotype	AR Profile	Total	H	M	E
7	P1	TET/ERY/STR/GEN/LZD/CIP/LVX	3	0	3	0
6	P2	TET/ERY/STR/LZD/AMP/PEN	2	0	0	2
5	P3	TET/ERY/STR/NIT/VAN	1	0	0	1
4	P4	TET/ERY/STR/GEN	13	13	0	0
	P5	TET/ERY/STR/VAN	1	0	0	1
	P6	TET/ERY/LZD/NIT	1	1	0	0
	P7	TET/ERY/CIP/LVX	1	0	1	0
3	P8	TET/ERY/GEN	3	3	0	0
	P9	TET/ERY/LZD	2	2	0	0
	P10	TET/ERY/STR	3	1	1	1

H: human; M: milk; E: environment; AMP: ampicillin; CIP: ciprofloxacin; ERY: erythromycin; GEN: high-level gentamicin synergy; LVX: levofloxacin; LZD: linezolid; NIT: nitrofurantoin; PEN: benzylpenicillin; STR: high-level streptomycin synergy; TET: tetracycline; VAN: vancomycin.

**Table 3 antibiotics-14-00380-t003:** A description of the farms.

Farm	Number of Lactating Cows	Farm Scale
A	105	Large
B	21	Small
C	49	Semi-commercial
D	80	Commercial
E	5	Small
F	32	Semi-commercial
G	110	Commercial
H	25	Small

## Data Availability

Data cannot be publicly available due to privacy and ethical restrictions.

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
