# Peer review of "One Health Approach: Antibiotic Resistance Among Enterococcal Isolates in Dairy Farms in Selangor"

_antibiotics, 2025, doi:10.3390/antibiotics14040380_

Round 1
Reviewer 1 Report
Comments and Suggestions for Authors
In this manuscript, the authors investigate the AR distribution among enterococcal isolates in dairy farms at Selangor. Although, the subject is of interest, the findings can be submitted as a short format paper/ report. Extensive english language editing is required to improve the readability through out the manuscript. Given that this paper only focused on enterococcal isolates, the authors should include limitations section in the manuscript. They do include one line in the discussion but it needs to be expanded. There is a lack of adequate conclusions from the antibiotic resistance trends observed. Also, the authors need to address the findings and correlation of MAR index to the scale of farm. The significance and the practical advantage of these findings need to be better communicated.
Comments on the Quality of English LanguageExtensive english editing required to improve the readability of manuscript.
Reviewer 2 Report
Comments and Suggestions for Authors
Review
ONE HEALTH APPROACH: ANTIBIOTIC RESISTANCE AMONG ENTEROCOCCAL ISOLATES IN DAIRY FARMS IN SELANGOR
The paper is based on the current global issue of AR and clearly illustrates the importance of a One Health approach that integrates human, animal and environmental health. The authors rightly point out the need to monitor AR in different sectors, while in the context of dairy production, the emphasis is not only on direct transmission between humans and animals, but also on the impact of the environment.
Material and methodology - The selection of farms for the study was carried out based on the list of registered DVS, which provides a representative picture only of establishments under official supervision. However, this approach may lead to the omission of farms with different antibiotic use, thereby limiting the generalizability of the findings. Therefore, it would be appropriate to include farms that are not registered in DVS in future research.
The authors should clarify the criteria on which they decided to use the MAR index, as the original publication they cite suggests that the values calculated by this index ambiguously determined the increased or decreased risk associated with resistance to the antibiotics used, which is also confirmed by the examples given by the author Krumperman.
The method used to aggregate the MAR index for entire farms (a / [b x c]) is not sufficiently substantiated or compared with common approaches. The MAR index results are presented without a detailed discussion of their impact on the overall risk profile assessment, which makes their transparent interpretation difficult.
The graphs shown in Figures 1 and 2 lack information on absolute values and percentage of isolates, which is crucial for a complete and clear interpretation of the results. Adding this data would significantly contribute to a better interpretation and overall overview of the graphical presentation.
The results show that all LRE isolates exhibit resistance to both tetracycline and erythromycin, which is a significant phenomenon. However, a more detailed analysis of the mechanisms that could explain this co-resistance is lacking. Without the use of molecular analyses and in-depth phenotypic evaluation, it is not possible to clearly determine whether this is a genetically determined phenomenon or a phenotypic adaptation. Additional molecular studies could clarify the causes of co-resistance, thereby contributing to a better understanding of resistance mechanisms and the potential risks associated with their spread.
In Table 2, the explanatory notes list two abbreviations: BENPEN for Benzylpenicillin and PEN for Benzyl-Penicillin, which suggests that it could be Penicillin G. It is necessary to clarify whether BENPEN is a commercial name or is it an error, as these are two different abbreviations for one antibiotic drug. Furthermore, in the explanatory notes you list more than 12 antibiotics that were tested. Could you please clarify this inconsistency?
For multidrug-resistant antibiotic resistance, percentages are presented (e.g. 5 % MDR and distribution by number of resistance classes), but these data may lead to misinterpretations if it is not explicitly stated how these percentages were calculated with respect to the total number of samples and whether differences in the size of individual groups were taken into account.
The discussion primarily focuses on the description of the results and their consistency with expectations arising from previous studies, but lacks deeper critical reflection on possible sources of error, data variability, or alternative interpretations of the findings.
Although the discussion mentions important antibiotics and their classification (VCIA, critically important antimicrobials), it does not provide a detailed justification of how the findings on resistance could influence clinical practice or policy-making, especially in the field of veterinary medicine. Moreover, although the One Health approach is mentioned, there is no specification of how the knowledge gained could contribute to a better integration between human, veterinary and environmental health. I would therefore have expected a more detailed analysis of possible preventive measures and monitoring programmes that would result from the data obtained.
Round 2
Reviewer 1 Report
Comments and Suggestions for Authors
The authors have addressed the comments.